# Preparation and Characterization of Novel 5Mg(OH)_2_·MgSO_4_·7H_2_O Whiskers

**DOI:** 10.3390/ma15228018

**Published:** 2022-11-14

**Authors:** Xinkuang Ning, Chengyou Wu, Hongdu Chen

**Affiliations:** 1School of Civil Engineering, Qinghai University, Xining 810016, China; 2Qinghai Provincial Key Laboratory of Energy-Saving Building Materials and Engineering Safety, Xining 810016, China

**Keywords:** basic magnesium-sulfate whisker, admixture, ammonia, low-temperature synthesis, thermal decomposition

## Abstract

Herein, novel monodisperse basic magnesium sulfate whiskers (5Mg(OH)_2_·MgSO_4_·7H_2_O) were prepared under low-temperature and atmospheric-pressure conditions, using the admixture sodium citrate. X-ray diffraction, Raman spectroscopy, scanning electron microscopy, energy-dispersive spectroscopy, transmission electron microscopy, selected area electron diffraction, thermogravimetric analysis, Fourier-transform infrared spectroscopy, and X-ray photoelectron spectroscopy were used to characterize the structure and morphology of the whisker products. The analysis results showed that the product was composed of high-purity basic magnesium sulfate whiskers. The lengths and diameters of the whiskers were 10–20 μm and 0.1–0.2 μm, respectively, and their aspect ratios were higher than 30. The formation mechanism of 5Mg(OH)_2_·MgSO_4_·7H_2_O involved direct assembly from the precursors without the formation of magnesium hydroxide for redissolution. High-purity MgO whiskers with smooth surfaces were prepared using the as-prepared whisker products via thermal decomposition. Thus, the findings of this study can provide technical support for the cost-effective industrial-scale preparation of basic magnesium-sulfate whiskers and associated whisker products.

## 1. Introduction

Inorganic whiskers are micro-nanoscale fiber materials with specific aspect ratios formed via single-crystal growth. Their mechanical strength is equal to the strength generated by the force between adjacent atoms, and its value is approximately equal to the theoretical value of a complete crystal without crystal defects. Inorganic whiskers have a wide range of applications, such as reinforcing materials for plastics, resins, rubber, and cementitious materials [1]. Common inorganic whiskers are primarily composed of materials such as silicon carbide, potassium titanate, aluminum borate, calcium sulfate, magnesium oxide, magnesium hydroxide, and magnesium salts. In recent years, an increasing number of studies have been conducted on inorganic whiskers worldwide, particularly magnesium whiskers [2,3,4]. Luo et al. found that boric-acid-ester-modified magnesium borate whiskers can significantly enhance the tensile and impact strength, crystallization rate, and nucleation activity of polypropylene composites [5]. Huang et al. prepared poly vinylidene fluoride scaffolds with MgO whiskers through selective laser sintering [6]; the tensile strength and elastic modulus increased by 52.53% and 29.31%, respectively. Wen et al. found that the crystallization rate and hydrophilicity of PLLA were improved by introducing surface-modified L-lactide to obtain grafted MgO whiskers (g-MgO whiskers) [7]. The g-MgO whisker/PLLA composites exhibited significantly higher strength, modulus, and toughness than the constituent materials. Li et al. incorporated magnesium-hydroxide micro-whiskers (mw-MHs) as super-reinforcements into epoxy resin (EP) [8]. The EP/mw-MH micro-composites outperformed the commonly used industrial magnesium hydroxide (ind-MH) in terms of fire safety at the same loading. More in-depth research on inorganic whiskers, which is of significant importance for the application and development of materials, is in progress.

The salt lakes in China are rich in magnesium resources, which are mainly distributed in the Qinghai–Tibet Plateau. The reserves of magnesium resources in the Qinghai Salt Lake amount to more than 4 billion tons, accounting for approximately 75% of the national reserves and 40% of the global reserves [9]. The potassium and lithium industries produce large amounts of magnesium salts. The production of each ton of potash fertilizer can generate 8–10 tons of high-magnesium old brine. The high-magnesium old brine discharged by potash fertilizer enterprises in the Chaerhan Salt Lake area is converted into approximately 6000 × 10^4^ tons of high-magnesium old brine (MgCl_2_·6H_2_O), and a large amount of magnesium-salt byproduct is re-discharged into the salt lake [10]. This results in significant wastage of resources and affects the sustainable production of potash fertilizers. Therefore, developing a low-cost preparation technology for magnesium-based whisker materials is essential for achieving the high-value utilization of magnesium resources in salt lakes. Basic magnesium-sulfate whiskers (BMSW), which are also known as magnesium-oxysulfate whiskers or magnesium hydroxide sulfate hydrate whiskers, are important inorganic-magnesium-salt whiskers, and their general structure can be expressed as xMg(OH)_2_·yMgSO_4_·zH_2_O (abbreviated as xyz-type BMSW). Currently, the 512-type and the 513-type are the most frequently studied BMSWs worldwide [11]. The preparation method of BMSW is simple, with high product purity. The Mg(OH)_2_ and MgO whiskers, prepared using BMSW as a precursor, have good dispersibility and high crystallinity. BMSW can also be added to composite materials, such as plastics, rubbers, and resins, as a material-reinforcement agent [12,13,14]. Chen et al. prepared natural rubber (NR)/BMSW composites by blending the BMSW modified by stearic acid or Si69 with NR latex [15]. The mechanical properties, anti-ultraviolet aging properties, flammability, and thermal stability of the composites were significantly improved by incorporating modified-BMSWs into NR. Zhang et al. added BMSW to magnesium silicate hydrate (M-S-H) cement, which reduced the porosity of the M-S-H cement mortars [16]. The optimal compressive and flexural strengths were obtained at BMSW contents between 3 and 4 wt%. Further, the crystal water contained in BMSW decomposes when heated, facilitating its application as a flame-retardant material [17,18,19,20]. The most common methods for preparing BMSWs are hydrothermal and microwave-hydrothermal methods [21,22,23]. Both methods involve reactions at high temperatures (150–200 °C) and high pressure (0.3–2 MPa), leading to potential safety hazards [24,25]. In addition, their equipment costs and energy consumption are relatively high; therefore, these two methods are not conducive to large-scale industrial production. Currently, few articles are available on the low-cost preparation of BMSWs. Fan et al. prepared BMSWs in NH_4_^+^–NH_3_ buffer system by a one-step method at atmospheric pressure, which reduced the reaction temperature to 75 °C [26]. Ma et al. developed a method to synthesize BMSWs using MgSO_4_·7H_2_O and MgO at reflux under atmospheric pressure, but this method required a preparation time of more than 25 h [27]. Wu et al. used Dezhou bittern as the raw material and sodium hydroxide and sodium sulfate as the precipitants, and prepared high-quality BMSWs by solar heating; however, the preparation time was more than 180 h [28]. Therefore, finding a low-cost and rapid method for preparing BMSWs is very important for the industrial development of magnesium-salt whiskers.

Wu et al. observed the microstructure of basic magnesium-sulfate cement and found a large number of needle-like 517 crystals (5Mg(OH)_2_·MgSO_4_·7H_2_O) [29]. When basic magnesium-sulfate cement is used as a seed material to strengthen the properties of other materials, the grinding method is adopted, which deteriorates the strengthening properties of the 517 phase. Therefore, the direct preparation of the 517-type BMSW as a reinforcing agent can have a more significant strengthening and toughening effect. However, no systematic report on the preparation of the 517-type BMSW has been published to date. Based on the formation mechanism of the 517 crystals in the MgO-MgSO_4_-H_2_O gelling system, the formation of the 517 phase requires the dissolution of MgO because the system provides the OH^−^ required for the formation of 517 crystals; however, the complete dissolution of MgO requires sufficient time. Therefore, the preparation of high-purity monodisperse 517-type BMSWs by directly using MgO under mild conditions may require a long time. Hence, monodisperse 517-type BMSWs were prepared by controlling the admixture sodium citrate (CS in short) and replacing MgO with ammonia aqueous solution in the liquid phase. The products were analyzed using X-ray diffraction (XRD), Raman spectroscopy (RS), scanning electron microscopy (SEM), energy-dispersive spectroscopy (EDS), transmission electron microscopy (TEM), selected area electron diffraction (SAED), X-ray photoelectron spectroscopy (XPS), thermogravimetry–differential scanning calorimetry (TG-DSC), Fourier-transform infrared (FTIR) spectroscopy, and other testing methods to characterize the structure and morphology of the whisker products. The 517-type BMSWs have more water molecules in their chemical formula, as well as superior flame-retardant and smoke-suppression functions, compared with other BMSWs. Further, high-purity MgO whiskers with smooth surfaces were preliminarily prepared via thermal decomposition using 517-type BMSWs.

## 2. Materials and Methods

### 2.1. Reagents

Ammonia aqueous solution ([NH_3_·H_2_O], analytical reagent, concentration 25–28%) was purchased from Shanghai McLean Biochemical Technology Co., Ltd. The specific concentration measured by titration with standard hydrochloric acid was 26.05%. Magnesium sulfate heptahydrate (MgSO_4_·7H_2_O) and admixture, analytically pure sodium citrate (Na_3_C_6_H_5_O_7_·2H_2_O), were purchased from Tianjin Zhiyuan Chemical Reagent Co., Ltd. (Tianjin, China). Sodium dodecyl sulfate ((C_12_H_25_SO_4_Na), analytical grade) was purchased from Tianjin Kaitong Chemical Reagent Co., Ltd. (Tianjin, China).

### 2.2. Preparation of 5Mg(OH)_2_·MgSO_4_·7H_2_O Whiskers

Sodium citrate (0.8% (0.3 g) per mass of MgSO_4_) was added to a 20% (by weight) magnesium sulfate solution (180 mL) while stirring at a speed of 500 rpm. After the dissolution of sodium citrate, 19.61 g of ammonia aqueous solution was added dropwise at a speed of 3.0 mL/min. Thereafter, 517-type BMSWs were obtained by continuous shaking at 35 °C for 18 h, washed successively with water and ethanol, and dried at 40 °C to a constant weight (Figure 1).

The prepared whiskers were stirred in 500 mL of sodium dodecyl sulfate (K12) solution with a concentration of 0.15% for 20 min at a stirring speed of 500 rpm. The maximum temperature was 1000 °C, and the heating rate was 10 °C/min. The K12 solution can be recycled. After filtering, it was maintained for 120 min, and MgO whiskers were obtained.

### 2.3. Test Analysis and Characterization

The heating rate during calcination was 10 °C/min, and the holding time was 120 min. The prepared samples were ground into powder using an agate mortar and sieved through a 200-mesh sieve (D < 75 μm). The XRD (D/max-2500PC) with Cu-Kα radiation (λ = 0.15419 nm) was performed at an accelerating voltage of 30 kV (2θ = 5°–70°, scan rate = 8°/min), and the obtained XRD patterns were analyzed. Raman spectroscopy (LabRAM HR Evolution) was performed in the spectral range of 50–1400 cm^−1^ with a focal length of 800 mm, and the spectrometer was equipped with a 532-nanometer laser with a laser power of <1 mV. A scanning electron microscope (COXEM EM-30) was used to characterize the microstructure of the gold-plated samples. The magnesium oxide whiskers were observed using a high-magnification electron microscope to analyze the microstructures (JSM-6610LV). A resolution of 4.0 nm was used, and the test voltage was 15 kV. The element type and content of the product were analyzed using an energy-dispersive spectrometer (X-ACT Compact 30) integrated into the SEM equipment. The internal structure of the samples was analyzed via TEM (FEI Talos F200X G2) and SAED. The prepared samples were ground into powder using an agate mortar and sieved through a 200-mesh sieve (D < 75 μm), and a microcomputer differential thermal balance (HCT-1) was used to conduct the thermal analysis of the samples. The testing was conducted under a nitrogen atmosphere in the temperature range of 25–1050 °C, with a heating rate of 10 °C/min. The samples were qualitatively and structurally analyzed using an FTIR Spectrometer (Nicolet 6700). The XPS measurements were performed using an X-ray photoelectron spectrometer (Thermo Fisher ESCALAB Xi^+^) to analyze the elements and valence states of the samples. A muffle furnace (BW-GWL-1200 °C) was used to prepare magnesium-oxide whiskers.

## 3. Results

### 3.1. Preparation

#### 3.1.1. Effects of CS Addition on the Preparation of 517-Type BMSW

Figure 2 shows the XRD patterns of the products with 0 and 0.8 wt% CS, which were compared with the standard diffraction patterns. In the absence of CS, the diffraction patterns of the product were consistent with the standard diffraction pattern of Mg(OH)_2_ (PDF#97-009-5475) [8]. The product showed five strong diffraction peaks at 2θ = 18.53°, 37.98°, 50.79°, 58.67°, and 62.12°, which were attributed to the (0,0,1), (1,0,1), (1,0,2), (1,1,0), and (1,1,1) planes, respectively. At 0.8 wt% CS, the product diffraction peaks were consistent with the standard diffraction pattern of phase 517 (PDF#97-042-5847) [30], and other impurity peaks were not observed. The product showed five strong diffraction peaks at 2θ = 9.44°, 17.80°, 30.90°, 37.34°, and 45.21°, which were attributed to the (1,0,−1), (2,0,0), (0,2,2), (2,2,2), and (4,0,−6) planes, respectively. These results suggest that in the absence of the admixture CS, the only reaction that occurred in the solution was that between Mg^2+^ and OH^−^ to generate Mg(OH)_2_.

The Raman spectroscopy results of the products with 0 and 0.8 wt% of CS (Figure 3) were consistent with the XRD-analysis results. In the case of the CS addition, intensive sulfate vibrations (symmetric stretching vibrations) were detected in the sample at 988 cm^−1^. The symmetric and asymmetric stretching vibrations of the sulfate were observed at 100–500 cm^−1^, but these vibrations were indistinguishable from the stretching vibrations of the Mg–O octahedra. Without the addition of CS, the Raman spectra of the samples showed sharp bands of the stretching vibrations of Mg–O at 279 and 444 cm^−1^; this was in agreement with the results of previous studies on Mg(OH)_2_ [31].

According to Figure 4, at 0 wt% CS, the product was composed of Mg(OH)_2_ particles. The SEM image of the product with a CS content of 0.8 wt% shows the smooth surface and neat morphology of the 517-type BMSWs with high purity.

According to previous reports [32,33], the formation of 517-type BMSWs requires an appropriate amount of admixture, such as CS; the mechanism of action of citrate in the reaction is shown in Figure 5. First, the ammonia aqueous solution was dropped into the solution, which was rapidly ionized to generate NH_4_^+^ and OH^−^. Subsequently, the pH value of the solution increased, and the OH^−^ reacted with the Mg^2+^ and H_2_O molecules to form Mg-O_6_ octahedrons with a sandwich structure and a specific interlayer spacing. With no citrate ions (CA^n−^) or very low contents in solution, only OH^−^ ions with a small radius could enter the Mg-O_6_ octahedral space to form Mg(OH)_2_ because the Mg-O_6_ octahedral interlayer spacing was very small (4.77 Å). After the addition of the citrate ions, owing to their strong complexation, they supported the Mg-O_6_ octahedral skeleton to ensure the entry of the aqueous sulfate, thereby generating the 517-type BMSW. The citrate in the liquid phase acted as a soft template during the formation of the 517-type BMSW. 

Chemical equations:(1)$${\mathrm{NH}}_{3}{\cdot H}_{2}O\to {\mathrm{NH}}_{4}^{+}{+\mathrm{OH}}^{-}$$
(2)$${\mathrm{Mg}}^{2+}{+\mathrm{OH}}^{-}{+\mathrm{nH}}_{2}O\to {\mathrm{Mg}(H}_{2}{O)}_{n}{\left(\mathrm{OH}\right)}^{+}$$

When no admixture cs is added:(3)$${\mathrm{Mg}(H}_{2}{O)}_{n}{\left(\mathrm{OH}\right)}^{+}{+\mathrm{OH}}^{-}\to {\mathrm{Mg}(\mathrm{OH})}_{2}{+\mathrm{nH}}_{2}O$$

When an appropriate amount of admixture CS is added:(4)$${\mathrm{Mg}(H}_{2}{O)}_{n}{\left(\mathrm{OH}\right)}^{+}{+\mathrm{CA}}^{n-}\to {\left[{\mathrm{CA}}^{n-}\to \mathrm{Mg}{\left(H_{2}O\right)}_{n-1}\left(\mathrm{OH}\right)\right]}^{+}{+H}_{2}O$$
(5)$${\mathrm{SO}}_{4}^{2-}+{\left[{\mathrm{CA}}^{n-}\to \mathrm{Mg}{\left(H_{2}O\right)}_{n-1}\left(\mathrm{OH}\right)\right]}^{+}{+5\mathrm{OH}}^{-}\to {5\mathrm{Mg}(\mathrm{OH})}_{2}\cdot {\mathrm{MgSO}}_{4}\cdot {7H}_{2}O+\left(4n-12\right)H_{2}O$$

#### 3.1.2. Effects of Reaction Time on Preparation of 517-Type BMSW

To further verify the formation mechanism of the 517-type BMSWs, the results of this study were compared with those of a study on the preparation of BMSWs via the hydrothermal method. Furthermore, the products obtained after the addition of 0.8 wt% of CS were analyzed. Figure 6 shows the XRD patterns of the products at different reaction times obtained by tracking the product after the addition of 0.8 wt% CS. As shown in Figure 6, the product did not crystallize before the reaction time of 4 h, at which point, a weak characteristic peak of phase 517 appeared. The intensity of the peak for phase 517 gradually increased with time. All the diffraction peaks could be indexed to the orthorhombic structure of 5Mg(OH)_2_·MgSO_4_·7H_2_O when the time exceeded 6 h. Additionally, no peaks corresponding to other impurities, such as magnesium hydroxide, were observed in the XRD pattern. This indicates that the formation mechanism in this study was different from that of the hydrothermal preparation of the 512-type and 513-type BMSWs, wherein the dissolution and regeneration of magnesium hydroxide are not required, and the structure directly self-assembles to form 517-type BMSWs [34]. 

The chemical reaction for the preparation of 5Mg(OH)_2_·MgSO_4_·7H_2_O is presented in Equation (6).
(6)$${6\mathrm{Mg}}^{2+}{+10\mathrm{OH}}^{-}{+\mathrm{SO}}_{4}^{2-}{+7H}_{2}O\to {5\mathrm{Mg}(\mathrm{OH})}_{2}\cdot {\mathrm{MgSO}}_{4}\cdot {7H}_{2}O\;$$

Based on Equation (6), the OH^−^ ion conversion rate (R_OH_) was calculated according to the following formula:$$R_{\mathrm{OH}\;}=\frac{m_{517}\times M_{\mathrm{NH}3}\times 10}{m_{\mathrm{NH}3}{\times n}_{\mathrm{NH}3}\times M_{517}}\times 100\%$$
where m_517_ represents the mass of the whisker product (g), *M*_517_ represents the molar mass of the 517-type BMSW (g/mol), m_NH3_ represents the mass of the ammonia aqueous solution in the raw material (g), n_NH3_ represents the concentration of the ammonia aqueous solution (%), and *M*_NH3_ represents the molar mass of the ammonia aqueous solution (g/mol).

Ammonia aqueous solution is weakly alkaline. Therefore, compared with the strong alkalinity of MgO, the conversion rate of the product was expected to be slower. However, the reaction rate was expected to be faster because of the liquid phase of the ammonia aqueous solution. From 1 to 3 h of reaction time, the conversion rate increased to 45%. The XRD patterns suggest that this was the generation stage of the precursor. From 3 to 6 h, the conversion rate was relatively stable. The conversion rate significantly increased from 6 to 9 h, reaching 52.5%, and gradually stabilized from 9 to 12 h. Finally, the conversion rate of the OH^−^ ions was 55.5%. The XRD pattern of the product indicates that the 517-type BMSWs were primarily generated from 6 to 9 h. Therefore, the optimal preparation time is 9 h. However, the reaction time can be controlled within 6–7 h by adding a small amount of 517 seed crystals.

### 3.2. Characterization

#### 3.2.1. EDS Analysis

To further determine the elemental composition of the whiskers, EDS was used to quantitatively analyze the surface of the whisker products. Figure 7 shows the EDS mapping of the whisker products; Mg, O, and S can be observed on the whisker-product surface with compositions of 28.9%, 66.6%, and 4.5%, respectively, and the atomic ratio of Mg to S was 6.42. The whiskers were dissolved in hydrochloric acid, and the ethylene-diamine-tetraacetic-acid-titration method was used to measure the atomic ratio of Mg to S in the solution. The calculated Mg-to-S atomic ratio was 6.03, which was close to that in the 517-type BMSW (6.00). 

#### 3.2.2. XPS Analysis

The XPS profile of the products with 0 wt% CS content in Figure 8a indicates that the whisker components were mainly composed of O and Mg, with binding energies of 531 and 1304 eV, respectively. The fitted peak of S could not be identified because of the very low intensity. Figure 8a shows that the Mg:O ratio of the products with 0 wt% CS increased, and Figure 8b shows that the elemental valence state in the sample was Mg(OH)_2_. Only a small amount of impurity, which appeared with the fitting peak of O at 532.4 eV, was observed and was presumed to have been due to a small fraction of the C–O bond [35]. Furthermore, the elemental content of the products with 0 wt% CS is listed in Table 1. The Mg:O ratio is 1:1.97.

The XPS profiles of the products with 0.8 wt% CS content (Figure 9a) indicate that the whisker components were mainly composed of O, S, and Mg with binding energies of 531, 169, and 1304 eV, respectively. In addition to the whisker-constituent elements (O, Mg, S), C was detected, which can be regarded as a contaminant. The peaks of the XPS profiles were quantitatively analyzed, and the element content of the products with 0.8 wt% CS is listed in Table 2. The ratio of Mg:S:O was 4.86:1:15.26, which was slightly lower than that obtained via EDS analysis.

Avantage (version 5.976, ThermoFisher Scientific) software was used for the deconvolution of the O 1s, Mg 1s, and S 2p narrow spectra. Figure 9 shows the high-resolution XPS profiles of O 1s (b), Mg 1s (c), and S 2p (d). The peak center of the Mg 1s fitting peak appeared at approximately 1304 eV, which indicates that only one elemental valence state of Mg-O was present in the whiskers. The high-resolution XPS profile of S 2p shows two fitting peaks centered at 169 and 170.3 eV, indicating that two chemical environments were present in the S element with one valence state. This was due to the spin-orbit splitting of element S in nuclear magnetic resonance [36]. These results indicate that the as-prepared products were BMSWs. Figure 9b presents the fitted O 1s spectrum, which shows the presence of two peaks at binding energies of 531.8 and 533.3 eV, corresponding to the O–H and S–O bonds, respectively [16]. Compared to the 513-type BMSWs, the O 1s profile of the 517-type BMSWs had a significantly higher proportion of O–H and S–O. Therefore, the prepared product was 517-type BMSW. 

#### 3.2.3. FTIR Analysis

Figure 10 depicts the FTIR spectra of the products with 0 and 0.8 wt% CS in the wavenumber region of 4000–400 cm^−1^. The strong vibrational peaks at 3718 and 3640 cm^−1^ are the asymmetric stretching vibration peaks of MgO–H, and the peaks at 3372 and 1645 cm^−1^ are the asymmetric peaks of HO–H in crystal water [29]. The stretching and bending vibration peaks reveal that the whiskers contained chemical water and crystal water, and based on the thermal decomposition law, various connecting forces acted between the water molecules in the whiskers and the magnesium–oxygen skeleton (that is, the crystallization was first lost at low temperatures). The strong peaks at 523, 656, and 1104 cm^−1^ correspond to the stretching and bending vibration of SO_3_–O in sulfate. The FTIR analysis results are consistent with the structural characteristics of the 5Mg(OH)_2_·MgSO_4_·7H_2_O phase. Figure 10 shows the FTIR spectrum of the sample with 0 wt% CS content. The main component of the product was Mg(OH)_2_, as confirmed by the hydroxyl-vibration peaks at 3697 and 3647 cm^−1^. The absorption peaks at 3427, 1645, and 1126 cm^−1^ corresponded to small amounts of water from the crystallization and unwashed magnesium sulfate. The absorption peak at 1416 cm^−1^ may be attributed to the carbonation of the product surface to generate magnesium carbonate [30].

#### 3.2.4. TEM and SAED Analyses

To determine the structure of a single 517-type BMSW, TEM and SAED were used to analyze the crystal structure of the whisker product. The TEM image of the whisker product is shown in Figure 11a. The prepared whisker product was a dense structure with no pores inside the whisker, and the surface of the whisker was free of irregular and impure particles, with a diameter of approximately 0.1–0.2 μm. In the SAED pattern of the whisker, although the electron-diffraction spots were not very clear, a typical single-crystal diffraction spectrum was observed, which provides strong evidence that the 517-type BMSW possesses a single crystal structure. Figure 11b presents the SAED pattern of the whisker product. Two distinct crystal phases are not observable. Clear lattice fringes can be observed on the surface with the same interplanar spacing of 0.241 nm, corresponding to the (2,2,2) crystal plane.

#### 3.2.5. TG–DSC Analysis and Pyrolysis of Whiskers

Figure 12 shows the TG–DSC curves of the products with CS contents of 0 and 0.8 wt%. In Figure 12a,b, the product with 0.8 wt% CS loses 10.1% and 24.07% of weight at 73 and 132 °C, respectively. Three water molecules were removed at 73 °C, and four water molecules were removed at 132 °C to form the anhydrous magnesium-sulfate phase (5Mg(OH)_2_·MgSO_4_); the theoretical weight-loss values were 10.07% and 23.51%, respectively. The two values were close to those of the actual weight loss. The error was attributed to the probable presence of a small amount of unreacted precursor Mg–O_6_ octahedra in the sample and was also related to the manipulation error during the release of the sample.
(7)$${5\mathrm{Mg}(\mathrm{OH})}_{2}{\cdot \mathrm{MgSO}}_{4}{\cdot 7H}_{2}O\overset{50-100\;{}^{\circ}C\;}{\to }{5\mathrm{Mg}(\mathrm{OH})}_{2}{\cdot \mathrm{MgSO}}_{4}{\cdot 4H}_{2}{O+3H}_{2}O(g)↑$$
(8)$${5\mathrm{Mg}(\mathrm{OH})}_{2}{\cdot \mathrm{MgSO}}_{4}{\cdot 4H}_{2}O\overset{100-200\;{}^{\circ}C}{\to }{5\mathrm{Mg}(\mathrm{OH})}_{2}{\cdot \mathrm{MgSO}}_{4}{+4H}_{2}O(g)↑$$

The third decomposition peak of the product was located at 427 °C, corresponding to a weight loss of 40.49%. Considering the results of the weight-loss analysis, the decomposition reaction is a reaction in which anhydrous magnesium sulfate is decomposed into magnesium oxide and magnesium sulfate, and its theoretical weight loss was 40.3% [37].
(9)$${5\mathrm{Mg}(\mathrm{OH})}_{2}{\cdot \mathrm{MgSO}}_{4}\overset{400-450\;{}^{\circ}C}{\to }{5\mathrm{MgO}+\mathrm{MgSO}}_{4}{+5H}_{2}O(g)↑$$

The fourth decomposition peak of the product was located at 981 °C, corresponding to a weight loss of 54.62%. This was caused by the pyrolysis of the magnesium sulfate to generate magnesium oxide, SO_2_, and O_2_, while the theoretical weight loss of the 517-type BMSW was 55.04%, which was close to the actual weight-loss value.
(10)$${\mathrm{MgSO}}_{4}\overset{950-1000\;{}^{\circ}C}{\to }{2\mathrm{MgO}+2\mathrm{SO}}_{2}\left(g\right)↑{+O}_{2}\left(g\right)↑$$

Based on the literature [38], the pyrolysis temperature of Mg(OH)_2_ is approximately 350–400 °C, which is consistent with the DSC data of the product with 0 wt% CS. At a dosage of 0 wt%, the residual mass of the product was 63.85%, implying a weight loss of 36.15%, which was a significant deviation from the theoretical weight loss of the Mg(OH)_2_ (31.03%). This was probably because a small amount of crystal water in the product with a dosage of 0 wt% decomposed at 50–200 °C, which corresponded to a weight loss of 4.8%. According to the DSC–TG curve, when the additive CS was not added, the product formed was Mg(OH)_2_, containing a trace of crystal water.

According to the TG–DSC curve of the whisker product, the 517-type BMSW decomposed into MgO at 1000 °C. Based on this concept, the pyrolysis of the 517-type BMSW was attempted to prepare MgO whiskers with more extensive applications [39]. Figure 13 shows the XRD pattern, SEM image, and EDS spectrum of the MgO whisker. The ratio of Mg:O was 1:0.98. Clearly, the product was a high-purity MgO whisker with a smooth surface. The diameter of the whisker was 0.1–0.2 µm, and the length was more than 10–20 µm.

## 4. Conclusions

The following conclusions were drawn from this study:(1)Under low-temperature and atmospheric-pressure conditions, monodisperse 517-type BMSWs (structural formula 5Mg(OH)_2_·MgSO_4_·7H_2_O) were synthesized for the first time. The minimum preparation time was 6 h. In this study, for the first time, a method was developed for the rapid synthesis of BMSWs at ambient temperature and pressure. This method consumes less energy and is quicker than all the previous methods used for preparing BMSWs. This opens up a completely new pathway for the development of BMSWs.(2)The structure and morphology of the products with CS contents of 0 and 0.8 wt% were characterized via various techniques, including XRD, SEM, EDS, FTIR spectroscopy, TG–DSC, and XPS. The formation mechanism of the whiskers was analyzed. The as-prepared 517-type BMSWs exhibited high purity with lengths and diameters of 10–20 µm and 0.1–0.2 µm, respectively, and aspect ratios of higher than 50, which provides important technical support for future research on 517-type BMSWs. However, the recycling mechanism of the ammonia aqueous solution was not fully utilized herein. In future studies, ammonia gas can be distilled using the lime method, allowing for the ammonia aqueous solution to be reused.(3)Through thermal decomposition, high-purity MgO whiskers with smooth surfaces were pyrolyzed from the prepared whisker products, with lengths and diameters of 10–20 µm and 0.1–0.2 µm, respectively. The feasibility of preparing correlative whiskers from 517-type BMSWs was demonstrated.

## Figures and Tables

**Figure 1 materials-15-08018-f001:**
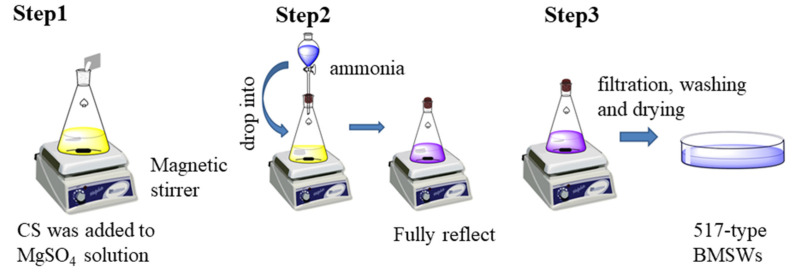
Preparation process of 517-type BMSWs.

**Figure 2 materials-15-08018-f002:**
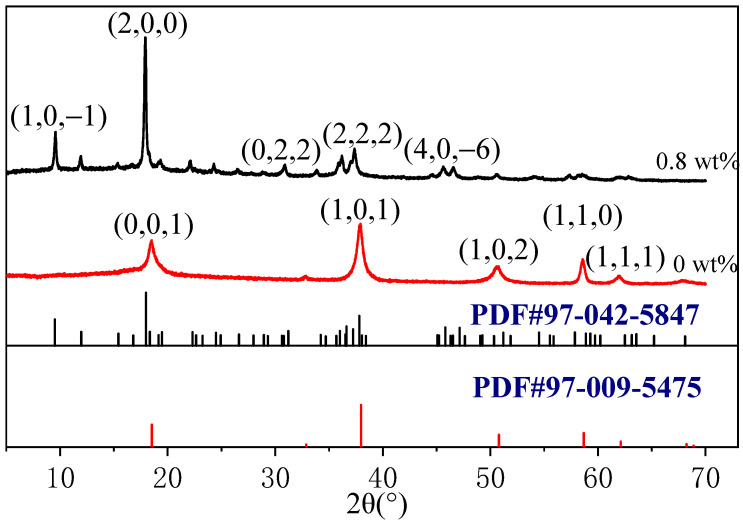
XRD patterns of products with 0 and 0.8 wt% of CS.

**Figure 3 materials-15-08018-f003:**
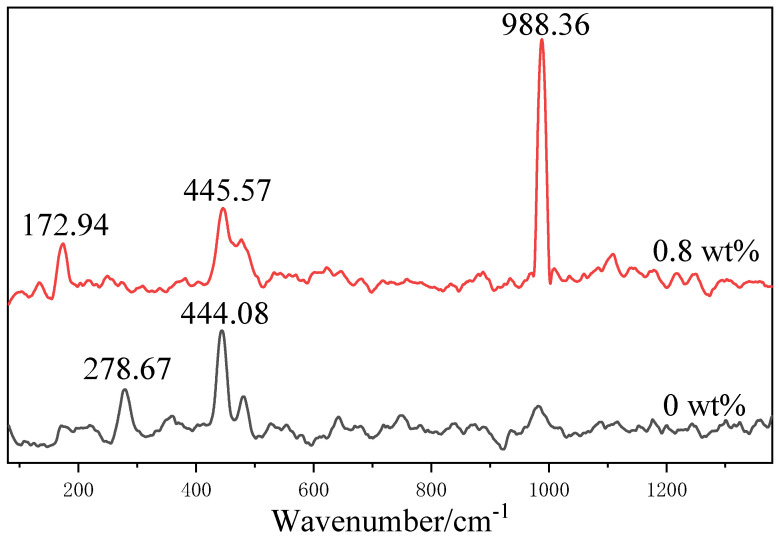
Raman spectra of products with 0 and 0.8 wt% of CS.

**Figure 4 materials-15-08018-f004:**
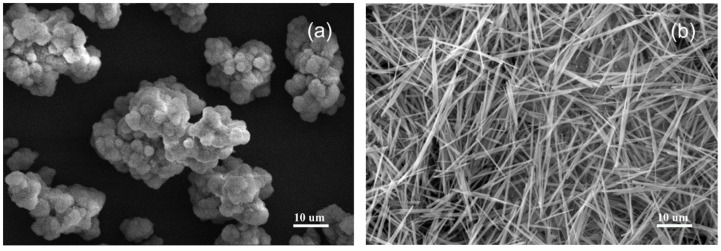
SEM images of the products with and without CS; (**a**) 0 wt%; (**b**) 0.8 wt%.

**Figure 5 materials-15-08018-f005:**
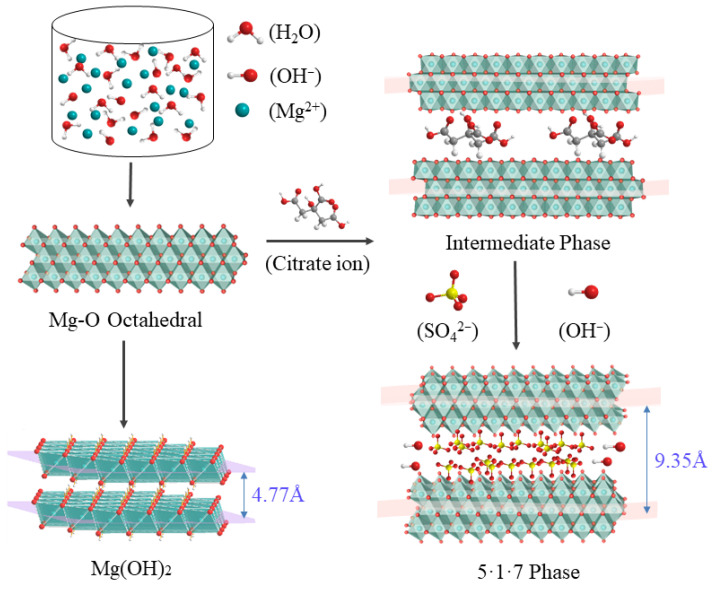
Schematic showing the formation mechanism of 517-type BMSW.

**Figure 6 materials-15-08018-f006:**
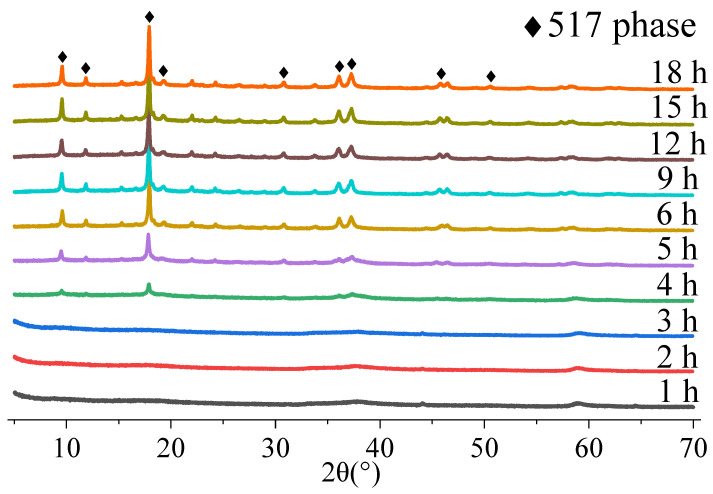
Tracking XRD patterns of samples with the addition of 0.8 wt% of CS.

**Figure 7 materials-15-08018-f007:**
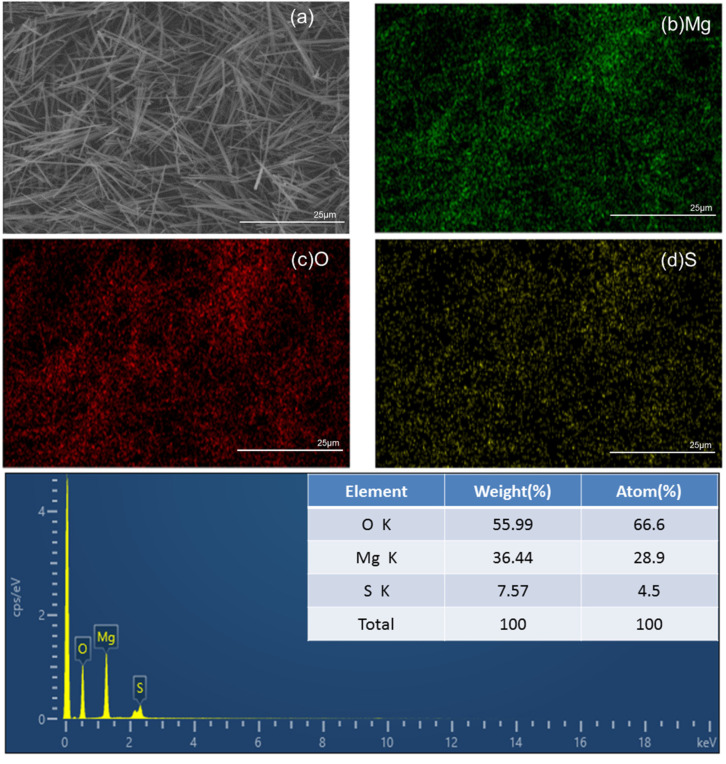
EDS mapping of whisker products. (**a**) is the original figure of figure (**b**–**d**).

**Figure 8 materials-15-08018-f008:**
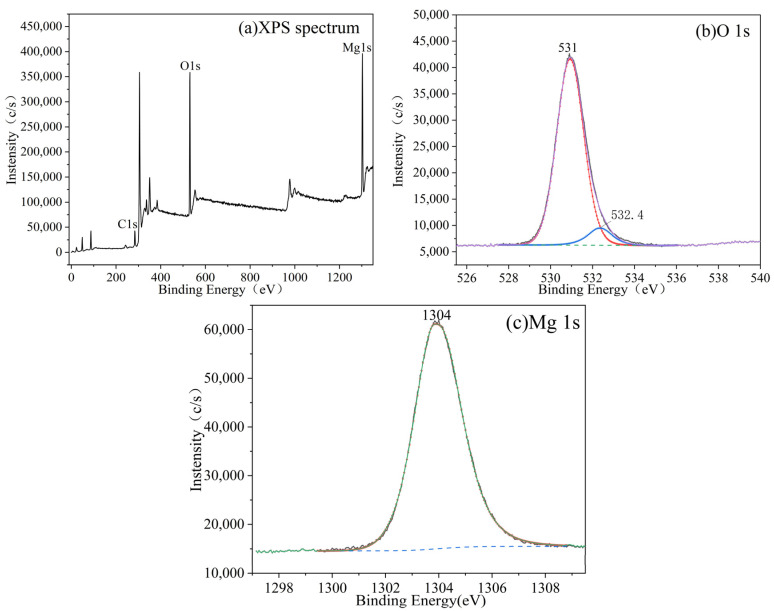
XPS profile of the products with 0 wt% CS (**a**), and high-resolution XPS profiles of O 1s (**b**) and Mg 1s (**c**).

**Figure 9 materials-15-08018-f009:**
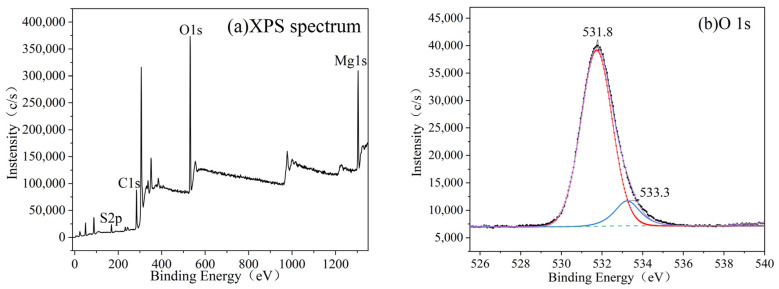
XPS profile of the products with 0.8 wt% CS (**a**), and high-resolution XPS profiles of O 1s (**b**), Mg 1s (**c**), and S 2p (**d**).

**Figure 10 materials-15-08018-f010:**
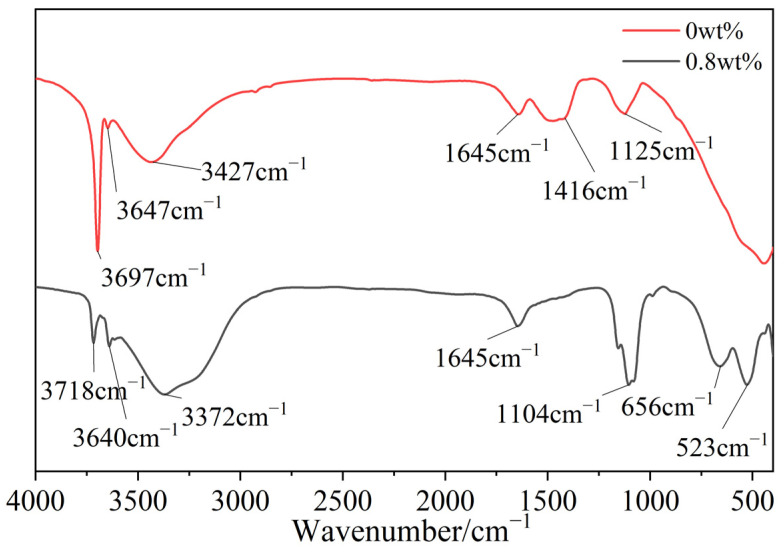
FTIR spectra of products with different CS contents.

**Figure 11 materials-15-08018-f011:**
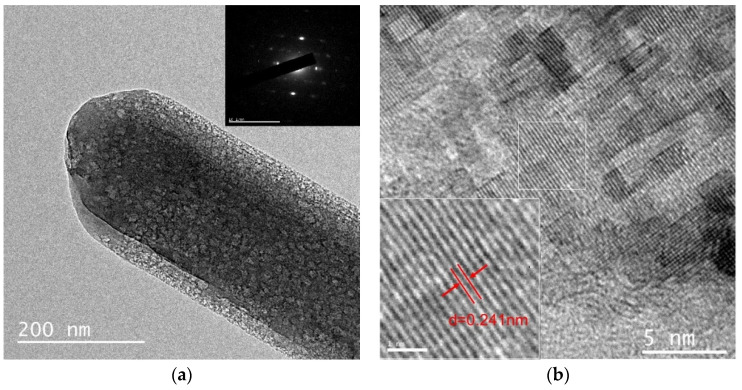
TEM images (**a**) and SAED pattern (**b**) of whisker products.

**Figure 12 materials-15-08018-f012:**
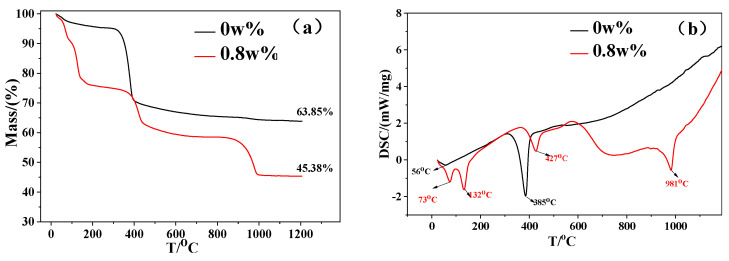
TG (**a**) and DSC (**b**) curves of each sample.

**Figure 13 materials-15-08018-f013:**
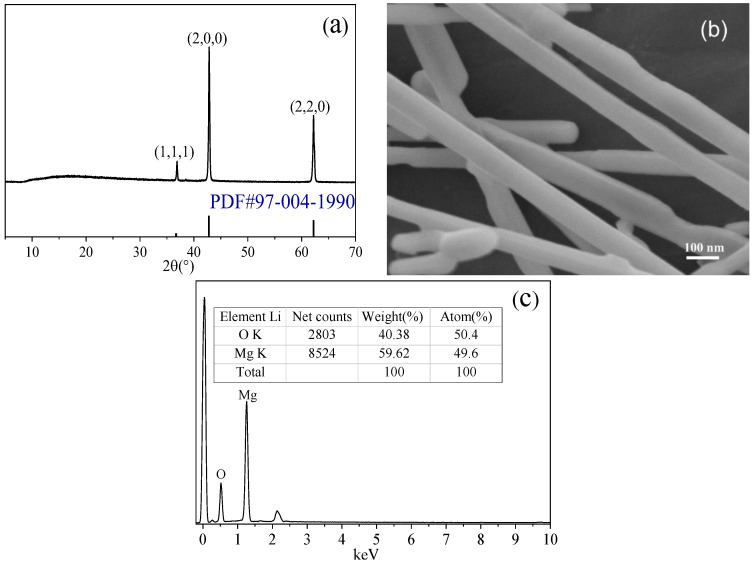
XRD pattern (**a**), SEM image (**b**), and corresponding EDS spectrum (**c**) of the MgO whisker.

**Table 1 materials-15-08018-t001:** Elemental content of the products of 0 wt% CS analyzed using XPS.

Element	Mg	O
Atomic%	33.69	66.31

**Table 2 materials-15-08018-t002:** Elemental content of the products of 0.8 wt% CS analyzed using XPS.

Element	Mg	O	S
Atomic%	23.02	72.24	4.74

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
