# Peer review of "Preparation and Characterization of Novel 5Mg(OH)2·MgSO4·7H2O Whiskers"

_materials, 2022, doi:10.3390/ma15228018_

Round 1
Reviewer 1 Report
The article is interesting, the investigation provides a study about the preparation and characterization of novel whiskers. Monodisperse 517-type BMSWs were prepared by controlling the admixture sodium citrate and replacing MgO with amonia wáter in the liquid pase. In this manuscript the samples were characterized by XRD, SEM-EDS, TEM, SAED, XPS, TG-DSC and FTIR.. The article could be published but there are some points that must be corrected before publication:
In Pag.3 and line 3. You must correct the following , he tensile strength…..
In Pag. 4 and line 3. You must correct the following, Is rssential for……
In Pag 11 and line 5. Why did you choose to work with 0.8 wt. % of CANA ?
In Pag 11 and line 8. Why after 4 h. products crystallize? In Fig.4
In Pag 15and Line 3. You must put binding energy of O, S, Mg, with respect to other works that have considered the same binding energies. For example,Effects of irradiation energy and nanoparticle concentrations on the structure and morphology of laser sintered magnesia with alumina and iron oxide nanoparticles, they considered Mg in 1304 eV
In Pag. 17. In Fig. 6 Why there was no shift in the Mg peak, since the Mg peak remained with the same binding energy, with respect to Figure 7.
In Fig. 6. Why there is no binding energy of Na?
In Fig 6 and 7. Why no peaks are identified in (a)?
Fig. 6. What are the two chemical environments which correspond to 169 and 170.3 eV? As a suggestion, you should put the starting material first (Fig. 7) and then the figure with the additions (Fig. 6).
When these questions are answered, the article will be published
Author Response
Response to Reviewer 1 Comments
Point 1: In Pag.3 and line 3. You must correct the following , he tensile strength…..
Response 1: Thank you for the valuable suggestion. We have incorporated the suggested revisions in the manuscript.
Point 2: In Pag. 4 and line 3. You must correct the following, Is rssential for……
Response 2: According to the suggestion, we have amended this question in the article.
Point 3: In Pag 11 and line 5. Why did you choose to work with 0.8 wt. % of CANA ?
Response 3: Thank you for the query. The sample with 0.8 w% sodium citrate was used for the follow-up experiment because it was found that 0.8 w% sodium citrate was the most economical preparation condition. The follow-up effect of other sodium citrate supplementation on the reaction needs to be further studied.
Point 4: In Pag 11 and line 8. Why after 4 h. products crystallize? In Fig.4.
Response 4: According to the XRD pattern, the product did not have any characteristic peak before 4 hours. When the reaction time was 4h, a weak characteristic peak of 517 phase appeared. Therefore, it is judged that the crystal nuclei of phase 517 are generated and grown at this time.
Point 5: In Pag 15and Line 3. You must put binding energy of O, S, Mg, with respect to other works that have considered the same binding energies. For example, Effects of irradiation energy and nanoparticle concentrations on the structure and morphology of laser sintered magnesia with alumina and iron oxide nanoparticles, they considered Mg in 1304 eV
Response 5: When Avantage software is used to analyze the full spectrum of XPS, the binding energies of O, S and Mg can be obtained directly, which are direct data that can be obtained through software analysis. In addition, the detailed analysis of each element in the following will be justified by citing the corresponding literature, which we believe can meet your requirements.
Point 6: In Pag. 17. In Fig. 6 Why there was no shift in the Mg peak, since the Mg peak remained with the same binding energy, with respect to Figure 7.
Response 6: Thank you for the query. Since the Mg of the two samples are in the same environment, the binding energy of Mg in the two XPS spectra should be the same. However, due to instrument error and other factors, the binding energy can not be completely consistent, and there will be little deviation.
Point 7: In Fig. 6. Why there is no binding energy of Na?
Response 7: The action of sodium citrate in the reaction belongs to admixture. During preparation, the sample is filtered and washed. Sodium citrate is dissolved in water and filtered, so Na element will not be measured in the sample.
Point 8: In Fig 6 and 7. Why no peaks are identified in (a)?
Response 8: In the XPS spectrum, many peaks are repeated peaks of the detected elements. This is because the spectrum peaks appearing in the photoelectron spectrum are the emission of certain characteristic energy electrons in the atom, and the electrons in each orbit are detected. For example, the peak at 300 eV is Mg 2p.
Point 9: Fig. 6. What are the two chemical environments which correspond to 169 and 170.3 eV? As a suggestion, you should put the starting material first (Fig. 7) and then the figure with the additions (Fig. 6).
Response 9: This is because of the two peaks produced by the asymmetric vibration of S element. Thank you for your valuable suggestions, we have made changes in the article.
Reviewer 2 Report
The article titled “Preparation and Characterization of Novel 5Mg (OH)2·MgSO4·7H2O Whiskers” in which the author subjected novel monodisperse basic magnesium sulfate whiskers (5Mg(OH)2·MgSO4·7H2O) were prepared under low temperature and atmospheric pressure conditions, using the admixture sodium citrate. Herein, novel monodisperse basic magnesium sulfate whiskers (5Mg(OH)2·MgSO4·7H2O) were prepared under low temperature and atmospheric pressure conditions, using the admixture sodium citrate. The research is interesting, but lacks of some major information. It is recommended to accept the article subjected to following major revisions:
1. In introduction section, author should discuss all possible strategies for low cost preparation technology of magnesium-based whisker with references.
2. In materials and methodology, draw an attractive methodology figure showing all methodology steps.
3. Author should mention the scan rate of samples in XRD characterization. In XRD analysis, 0.8 wt% showed multiple intermediate peaks which are not negligible. After CANA, elaborate all these peaks, also show their formation causes. Do not ignore any peak.
4. Author’s XRD discussion is not satisfied. Author must characterize the samples with Raman spectroscopy to assemble the XRD results further. Please discuss the explanation thoroughly.
5. In figure 2 SEM image, scale bar is not visible. Please make it visible for justification.
6. In FTIR, 0 wt% sample showed absorption peak at 1416, but 0.8 wt% have absence of this peak. How is this possible? Can you please check? Also show all hydroxyl peaks in the samples.
7. In TG curve, during 50– 200 ℃ what expected compounds, molecules are decomposed other than water? Mention all.
8. References in results and discussion part are very less. Conclusion is not comprehensive, improve it.
9. Overall English should be improved.
Author Response
Response to Reviewer 2 Comments
Point 1: In introduction section, author should discuss all possible strategies for low cost preparation technology of magnesium-based whisker with references.
Response 1: Thank you for the valuable suggestion. We have incorporated the suggested revisions in the manuscript.
Point 2: In materials and methodology, draw an attractive methodology figure showing all methodology steps.
Response 2: We have prepared a suitable schematic and added it to the Materials and methods section of the revised manuscript.
Point 3: Author should mention the scan rate of samples in XRD characterization. In XRD analysis, 0.8 wt% showed multiple intermediate peaks which are not negligible. After CANA, elaborate all these peaks, also show their formation causes. Do not ignore any peak.
Response 3: The scan rates of the samples in XRD characterization are 8 °/min. We have added this information in the Test analysis and characterization (Section 2.3) of the revised manuscript. We have modified the XRD plot as suggested by the reviewer and re-imported the PDF card, maintaining all values for l > 1 in the card. The modified XRD plot confirms that all intermediate peaks belong to the 517 phase.
Point 4: Author’s XRD discussion is not satisfied. Author must characterize the samples with Raman spectroscopy to assemble the XRD results further. Please discuss the explanation thoroughly.
Response 4: We supplemented the Raman spectral data of the samples and reinterpreted the analysis in conjunction with the XRD results.
Point 5: In figure 2 SEM image, scale bar is not visible. Please make it visible for justification.
Response 5: Thank you for the suggestion. Accordingly, we have revised the SEM images and included the information, such as the scale bar and magnification, below the image with better clarity.
Point 6: In FTIR, 0 wt% sample showed absorption peak at 1416, but 0.8 wt% have absence of this peak. How is this possible? Can you please check? Also show all hydroxyl peaks in the samples.
Response 6: We repeated the FTIR analysis of the sample, and the results obtained were the same as those obtained before. According to references [29,30], 517 phase does not exhibit a very obvious peak at 1416 cm-1, whereas the FTIR spectrum of magnesium carbonate shows an obvious peak at 1416 cm-1. According to these factors, the main product of the 0 wt% sample is considered to be magnesium hydroxide, which is easily carbonized by CO2 in air. Therefore, the peak at 1416 cm-1 corresponds to magnesium carbonate generated by the carbonization of 0 wt% sample. We have added this discussion in the revised manuscript and added the hydroxyl peaks.
Point 7: In TG curve, during 50–200 ℃ what expected compounds, molecules are decomposed other than water? Mention all.
Response 7: At 200 ℃, the sample weight loss was 24.07%, which was different from the theoretical weight loss of 23.51%. We presume that this can be attributed to a small amount of unreacted precursor Mg-O6 octahedra which might be present, and also to the manipulation error during sample release. We have added these explanation in the revised manuscript.
Point 8: References in results and discussion part are very less. Conclusion is not comprehensive, improve it.
Response 8: We have added a few relevant references to the Results and discussion section, strictly following the citation guidelines. Furthermore, we have also revised the conclusions.
Point 9: Overall English should be improved.
Response 9: According to the suggestion, we have carefully gone through the entire manuscript and further improved the English of the manuscript.
Reviewer 3 Report
Please discuss the potential application of synthesized material. The purpose of the studies is not clear, especially when it comes to the discussion and conclusions.
The obtained data are poorly compared with the literature - please compare the results with the data for similar compounds, the effectiveness of the proposed method, structure, etc.
This work of full of editorial mistakes (many of them in the whole manuscript) e.g.: n (180mL) while stirring at a speed of 500 r/min, where there should be a space between the number and unit, and rpm should be used - please correct them.
Author Response
Response to Reviewer 3 Comments
Point 1: Please discuss the potential application of synthesized material. The purpose of the studies is not clear, especially when it comes to the discussion and conclusions.
Response 1: Thank you for the suggestion. We have added a number of articles on the applications of basic magnesium sulfate whiskers and have improved the conclusions accordingly, further discussing the potential applications of the material. Because the chemical composition of basic magnesium sulfate whiskers is roughly the same, their applications are similar.
Point 2: The obtained data are poorly compared with the literature - please compare the results with the data for similar compounds, the effectiveness of the proposed method, structure, etc.
Response 2: We have improved the content of the article. Because we prepared a new material, which has not been studied and characterized before, there may be a few unavoidable deviations from the data in the literature.
Point 3: This work of full of editorial mistakes (many of them in the whole manuscript) e.g.: n (180mL) while stirring at a speed of 500 r/min, where there should be a space between the number and unit, and rpm should be used - please correct them.
Response 3: We are sorry for this error. We have made corrections to the overall editing errors and formatting of the article.
Reviewer 4 Report
The manuscript reports on a low-temperature synthesis of monodisperse 5Mg(OH)2·MgSO4·H2O whiskers. The paper topic is within the scope of materials journal, but the aim and the novelty of the paper is weakly emphasised. There is no data on any functional properties of the obtained materials.- In the Introduction section, the claim on the increasing number of studies worldwide does not agree with a single reference. Please add data in the introduction.
- The structure of the work is not entirely clear: the Results and Discussion section is mentioned twice (under numbers 3 and 3.2).
- The designation of sodium citrate as CANA is confusing.
- How were the samples ground into powder? How was the particle size in the powder determined?
- According to Figure 3, the resulting 517 phase is a layered one. Please specify the composition of the interlayer space. What is the evidence that the resulting 517 phase is layered? What are the anion-exchange properties of the resulting phase?
- How was the amount of water in the resulting 517 phase determined?
- What is the significance of the OH-ion conversion rate?
- Please specify how exactly the mechanism of formation of the 517 phase differs from the formation of 512 and 513?
- Why was the BMW 517-type annealing product coated with sodium dodecyl sulfate? It seems that some of the information has been lost. The section on the characteristics of the obtained MgO requires a discussion of its real chemical composition.
- The design of microphotographs requires uniformity: an image with a scale bar. The rest of the information (detector type, working distance, magnification, etc.) is not needed in the presentation of the results, but is needed in the Experimental.
Author Response
Response to Reviewer 4 Comments
Point 1: In the Introduction section, the claim on the increasing number of studies worldwide does not agree with a single reference. Please add data in the introduction.
Response 1: Thank you for the valuable suggestion. Based on the suggestion, we have added two new references on the application of magnesium salt whiskers.
Point 2: The structure of the work is not entirely clear: the Results and Discussion section is mentioned twice (under numbers 3 and 3.2).
Response 2: We have already rectified this issue in the manuscript.
Point 3: The designation of sodium citrate as CANA is confusing.
Response 3: To address this issue, we have changed the abbreviation of sodium citrate from CANA to CS.
Point 4: How were the samples ground into powder? How was the particle size in the powder determined?
Response 4: The prepared samples were ground into powder using an agate mortar and sieved through a 200-mesh size (D < 75 μm). We have included this information in the revised manuscript.
Point 5: According to Figure 3, the resulting 517 phase is a layered one. Please specify the composition of the interlayer space. What is the evidence that the resulting 517 phase is layered? What are the anion-exchange properties of the resulting phase?
Response 5: The composition of the interlayer structure is explained in detail in the article by T. Runčevski (Ref. [32]) under Section (2) Crystal Structure Description in Results and Discussion. In addition, the article by Miao (Ref. [33]) also confirms the layered structure of 517 phase. The results of the present study indicate that the citrate roots hold the skeleton at a certain spacing, allowing the hydroxide and sulfate ions to enter and react, and the nuclei to grow. Further studies are required to determine whether the citrate root is exchanged out or not.
Point 6: How was the amount of water in the resulting 517 phase determined?
Response 6: Based on the TG–DSC data of high-purity 517-type BMSW (high purity can be confirmed using other test methods), the weight loss of the sample at 200 °C was determined to be approximately 23.5%; the amount of water in the 517 phase can be calculated from this result.
Point 7: What is the significance of the OH-ion conversion rate?
Response 7: Thank you for the query. The OH-ion conversion rate is used to demonstrate the utilization of ammonia.
Point 8: Please specify how exactly the mechanism of formation of the 517 phase differs from the formation of 512 and 513?
Response 8: The most critical difference is that the formation of both 512-type and 513-type BMSW requires the dissolution and subsequent generation of Mg(OH)2. During this process, a high temperature and high pressure reaction environment is usually required to promote the dissolution of Mg(OH)2 (i.e., high energy consumption). Therefore, usually the hydrothermal method is used. In contrast, 517-type BMSW is rapidly generated from the precursor Mg-O6 octahedra. Subsequently, the skeleton structure is held by citrate ions, and hydroxide and sulfate ions enter the skeleton for direct reaction. This does not require a high-temperature and high-pressure reaction environment.
Point 9: Why was the BMW 517-type annealing product coated with sodium dodecyl sulfate? It seems that some of the information has been lost. The section on the characteristics of the obtained MgO requires a discussion of its real chemical composition.
Response 9: This choice is based on the cost factor during the industrial production of MgO; the cost of sodium dodecyl sulfate is less compared to that of alcohol because of its recyclability. Here, we demonstrated that soaking with sodium dodecyl sulfate solution can also yield optimal dispersion of the whisker product. We have added the missing details of the process in the manuscript, and also added a discussion on the chemical composition of magnesium oxide.
Point 10: The design of microphotographs requires uniformity: an image with a scale bar. The rest of the information (detector type, working distance, magnification, etc.) is not needed in the presentation of the results, but is needed in the Experimental.
Response 10: The microstructures of magnesium oxide whiskers were observed using a high magnification electron microscope. We have added the details of the model of the microscope and other relevant information to the materials and methods section of the revised manuscript.
Round 2
Reviewer 1 Report
Now, the manuscript has the suggested changes and the questions have been answered, so it can be approved.
Author Response
Response to Reviewer 1 Comments
Point 1: Now, the manuscript has the suggested changes and the questions have been answered, so it can be approved.
Response 1: Thank you very much for your recommendation.
Reviewer 2 Report
It is recommended to accept the article for publication.
Author Response
Response to Reviewer 2 Comments
Point 1: It is recommended to accept the article for publication.
Response 1: Thank you very much for your recommendation.
Reviewer 3 Report
Line 42 - Remove "Error! Reference source not found."
Line 16 - unify fonts
Fig. 1 - improve the quality of the image
Fig. 3 - add space between wavenumber and brackett
line 269 - ammonia aqueous solution - not ammonia water
Please compare deeper obtained results with the literature.
In my opinion, after these corrections, the article can be published.
Author Response
Response to Reviewer 3 Comments
Point 1: Line 42 - Remove "Error! Reference source not found.".
Response 1: Thank you for bringing this to our attention. All the formatting errors including those within the quotes have been removed.
Point 2: Line 16 - unify fonts.
Response 2: Thank you for bringing this to our attention, and the required changes have been made. The article has been formatted to comply with the journal guidelines.
Point 3: Fig. 1 - improve the quality of the image.
Response 3: We are sincerely apologetic that certain images in the article were in the wrong format, and thus, we have replaced the images with better-quality images.
Point 4: Fig. 3 - add space between wavenumber and brackett.
Response 4: Thank you for bringing this to our attention. We have made the required changes in the article. We have also checked and corrected similar problems in other images.
Point 5: line 269 - ammonia aqueous solution - not ammonia water.
Response 5: Thank you for your valuable comments, we have made the required changes in the article.
Point 6: Please compare deeper obtained results with the literature.
Response 6: Thank you for your suggestion. The experimental results and literature are compared in the conclusion
Reviewer 4 Report
The authors corrected most of my concerns, nevertheless some further changes must be considered. The uniformity of the designation of sodium citrate is poor. For example, in Figure 1, it is called SC, not CS, as the authors indicated. Please transfer the details on the soaking of 517-type whiskers in sodium dodecyl sulfate (response 9) to the experimental section. Figure 4 has remained unchanged, the quality of the images is poor, they contain unnecessary information. Some references are missing. Please also indicate clearly the possible practical applications of the material obtained.Author Response
Response to Reviewer 4 Comments
Point 1: The uniformity of the designation of sodium citrate is poor. For example, in Figure 1, it is called SC, not CS, as the authors indicated.
Response 1: We sincerely apologize for the nonuniform use of abbreviations. We have corrected it and have also checked the entire article for similar errors.
Point 2: Please transfer the details on the soaking of 517-type whiskers in sodium dodecyl sulfate (response 9) to the experimental section.
Response 2: Thanks for your suggestion. We have moved the experimental details to the experimental section.
Point 3: Figure 4 has remained unchanged, the quality of the images is poor, they contain unnecessary information.
Response 3: Thank you for bringing this to our attention. We have processed Figure 4 and removed all the unnecessary information.
Point 4: Some references are missing.
Response 4: Thank you for bringing this to our attention. We have corrected the formatting error that appeared throughout the article
Point 5: Please also indicate clearly the possible practical applications of the material obtained.
Response 5: Thank you for your suggestion. We have added the potential practical applications of 517-type BMSWs in the introduction section.